# Influence of Borehole Casing on Received Signals in Downhole Method

**Shi Yan [1], Yan Yu [2,\*], Wenjun Zheng [1], Jie Su [3,4] and Zhenghua Zhou [4]**

[1] Guangdong Provincial Key Laboratory of Geodynamics and Geohazards, School of Earth Sciences and Engineering, Sun Yat-sen University, Guangzhou 510275, China
[2] National Institute of Natural Hazards, Ministry of Emergency Management of China, Beijing 100085, China
[3] School of Civil Engineering and Architecture, Wuyi University, Jiangmen 529030, China
[4] Institute of Geotechnical Engineering, Nanjing Tech University, Nanjing 210009, China
[\*] Correspondence: yanyu@ninhm.ac.cn

**Abstract:** Borehole shrinkage and collapse are likely to occur when downhole testing is conducted in soft or loose sandy soils, resulting in testing interruption. To prevent this situation from occurring, installing casing in the borehole is a common approach. However, in actual testing, the quality of the signal obtained from measuring points within the depth of the casing is often not ideal, and there is still no clear and unified justification for the causes of interference generated by the casing. Therefore, the team attempt to investigate and elucidate the impact of casing through on-site experiments and numerical simulations. Firstly, on-site tests on the impact of different materials of casing on the wave velocity test utilizing the downhole method are conducted, the waveform characteristics of the measurement points inside the PVC casing and steel casing boreholes are analyzed, and the usability of the test results are evaluated. Next, the contact state between shallow soil and casing is changed, and its impact on the waveform characteristics of signal at different depth measurement points is analyzed. Then, the ABAQUS finite element software is utilized to establish a three-dimensional finite element model for wave velocity testing using the casing method, and the dynamic response of the measuring points on the casing wall inside the hole under surface excitation is solved. By numerically simulating different casing materials, the contact state between the casing and the hole wall, and the presence of low wave velocity filling soil around the casing, the variation patterns of the affected measurement point signals in the time and frequency domains are investigated. Furthermore, combined with the measured data, the impact characteristics of the casing on the results of the wave velocity testing using the downhole method are systematically explored. This research can provide some insights for the application and data interpretation of signals in the downhole methods of cased wells.

**Keywords:** downhole method; numerical simulation; casing effect; shear wave velocity; travel time; in-situ experiment



## 1. Introduction

The in-situ shear wave velocity profile of the site is a crucial geotechnical parameter for engineering sites [1–5], and the downhole method of shear wave testing is one of the most commonly used in-situ wave velocity testing methods for obtaining shear wave velocity under small strain on the site, which has advantages such as low cost, simple principle, and convenient operation [6–9]. At present, research on the downhole method testing mainly focuses on travel time calculation [10–12] and propagation distance calculation [13], mostly neglecting the analysis of the reliability of obtaining signals.

Many factors can affect the quality of signal acquisition in on-site testing. Currently, there is a consensus that the influencing factors include the contact state between the sleeper and the surface, the distance between the sleeper and the borehole, and etc. [14–16]. However, there is relatively little research on the impact of casing on the

acquisition of signals through the downhole method of wave velocity testing. When conducting downhole testing on weak sites or sites with loose sandy soil layers, it is easy to experience borehole shrinkage and collapse, which can engender testing interruption. To prevent this situation, lowering the casing during drilling is a typical practice. However, in actual testing, the quality of the signal acquired from the measuring points within the depth of the casing is often not ideal. Larkin and Taylor [17] compared the signals collected by drilling holes before and after casing placement under forward and reverse percussion and found that the phase of the signals gained by shallow and mid layer measurement points after casing placement was significantly disturbed. The interference components were defined as the torsional and bending waves generated by percussion propagating along the casing, which attenuated rapidly with depth. Crice [18] compared the signals obtained from drilling with casing in the presence and absence of well fluid at measurement points and found that the waveform of the signals obtained in the wet hole state was substantially disturbed. Crice believed that the interference component was a pressure pulse formed by the coupling of soil, casing, and liquid, which had the characteristics of fast wave velocity and slow attenuation along depth. Therefore, in deeper measurement points, it could be mistakenly identified as shear waves and cause interference. In fact, there is still no clear and unified justification for the reason of interference caused by the casing. Moreover, due to the expansion of the hole during the drilling process, the loose sand layer in the hole falling off, and the bridge plug phenomenon during the soil subsidence process, the contact state between the casing and the hole wall soil is complex, which can easily produce a cavity situation and impact the acquisition of signals. The analysis of the contact state between the casing and the hole wall on the testing site is time-consuming and labor-intensive; therefore, it is generally believed that the signal obtained in the depth of the casing in the downhole method of wave velocity testing is not reliable, and the test signal of the measurement point in the depth of the casing is discarded to avoid the error generated by the casing. However, with the extensive construction of infrastructure in China's delta plain areas and cross river and sea engineering projects, there are more and more opportunities for encountering soft soil sites in in-situ wave velocity testing. Improving drilling stability through casing has become a norm, and the casing is also getting deeper. Many technical personnel in on-site wave velocity testing choose to abandon testing the wave velocity at shallow positions with casing. However, estimating the corresponding soil layer wave velocity referring to the wave velocity data of the nearby boreholes will result in more errors. It is meaningful to clarify the impact caused by the casing during the drilling method testing and find ways to eliminate the impact.

This study mainly focused on the influence of casing on the wave velocity testing signal in the downhole method. Field experiments on the influence of different materials of casing on the wave velocity test of downhole method were first conducted, and the signal wave characteristics and the usability of test results in the borehole of PVC casing and that of steel casing were analyzed. Then, the contact state between shallow soil and casing was altered to analyze its effect on the signal waveform characteristics of measurement points at different depths. In geotechnical engineering, there is a lot of available software, such as FLAC, UDEC or ABAQUS. Considering that our research group has purchased the genuine ABAQUS software, then in this study, the geotechnical simulation software is used to analyze the displacement contour of the roadway excavation [19]; the ABAQUS finite element software was harnessed to establish a three-dimensional finite element model for wave velocity testing with the casing downhole method and simulate the dynamic response of the measurement points on the casing wall in the hole under surface excitation. Through numerically simulating different casing materials, casing and borehole wall contact state, casing surrounded by a low-velocity soil layer, and other working conditions, the affected measurement point signal changes in the time domain and frequency domain were analyzed, and combined with the measured data, the characteristics of the influence of casing on the downhole method wave velocity test were systematically explored.

## 2. Field Experiment Preparation and Experimental Model

### 2.1. Experimental Site and Test Apparatus

The experimental site was located on the North Campus of the College of Disaster Prevention and Technology in Sanhe City, situated in the pre-Yanshan plain area, mainly alluvial from the Chaobai River and the Ji Canal. The main body of the site is fluvial deposits with sound stratigraphic conditions. A sketch of the stratigraphic section is shown in Figure 1a [20]. Two boreholes with 117 mm diameter and about 10 m apart were arranged in the site for placing the casing of different materials. The casing length of the PVC cased borehole exceeded 30 m, and that of the steel cased borehole exceeded 24 m. The acquisition system selected for the wave velocity test field experiment of the downhole method with cased borehole was the SE2404PLUS series integrated engineering detector (see Figure 1b) produced by Beijing Jopeng Group, equipped with a single three-component airbag probe (see Figure 1c). The power system excited shear energy by manually striking the ground sleeper horizontally (see Figure 1d,e). To increase the coupling between the sleepers and the ground, the ground surface was scraped and iron blocks were placed on top of the sleepers.

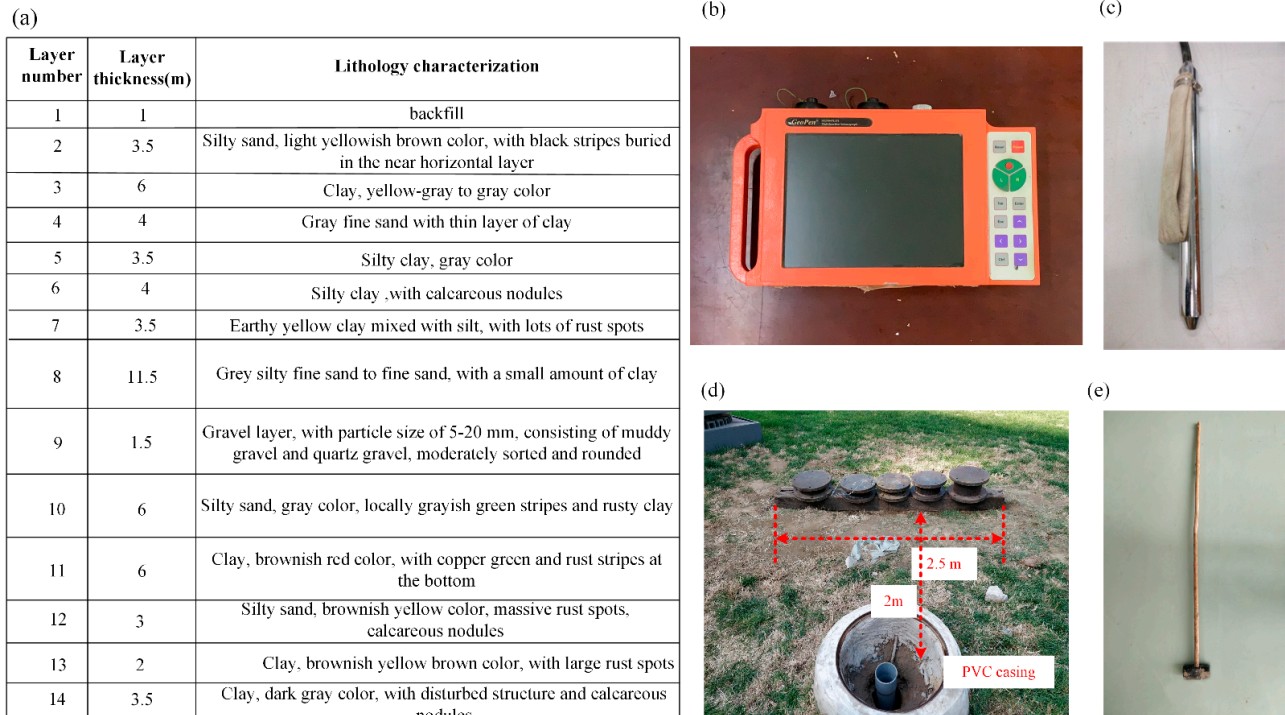

(a)

| Layer number | Layer thickness(m) | Lithology characterization |
|---|---|---|
| 1 | 1 | backfill |
| 2 | 3.5 | Silty sand, light yellowish brown color, with black stripes buried in the near horizontal layer |
| 3 | 6 | Clay, yellow-gray to gray color |
| 4 | 4 | Gray fine sand with thin layer of clay |
| 5 | 3.5 | Silty clay, gray color |
| 6 | 4 | Silty clay ,with calcareous nodules |
| 7 | 3.5 | Earthy yellow clay mixed with silt, with lots of rust spots |
| 8 | 11.5 | Grey silty fine sand to fine sand, with a small amount of clay |
| 9 | 1.5 | Gravel layer, with particle size of 5-20 mm, consisting of muddy gravel and quartz gravel, moderately sorted and rounded |
| 10 | 6 | Silty sand, gray color, locally grayish green stripes and rusty clay |
| 11 | 6 | Clay, brownish red color, with copper green and rust stripes at the bottom |
| 12 | 3 | Silty sand, brownish yellow color, massive rust spots, calcareous nodules |
| 13 | 2 | Clay, brownish yellow brown color, with large rust spots |
| 14 | 3.5 | Clay, dark gray color, with disturbed structure and calcareous nodules |

**Figure 1.** (**a**) Schematic diagram of soil profile of in-situ downhole method, (**b**–**e**) in-situ downhole method equipment.

### 2.2. Experimental Models and Test Method

#### 2.2.1. Experimental Models of Casing Borehole with Different Materials

In the field experiment, PVC casing and steel casing were installed in two boreholes to develop two experimental models of casing, and the casing was in close contact with the hole wall at this time. In order to strengthen the connection between the casing and the borehole wall, the diameter of the borehole should be slightly smaller than that of the casing during the excavation process. After the casing was established, sand was poured into the void between the surface casing and the hole wall. A three-component probe was placed in the borehole with casing and the signal of each measurement point was recorded by pulling the probe from the bottom up. The distance between the vertical center of the percussive sleeper and the center of the casing hole was 2 m. The test depths of the two casing holes with casing were 4 m to 22 m, and the spacing between the measuring points was 2 m. In an effort to enhance the reliability of signal acquisition, the same measuring point was

struck several times and the signal observed. The sampling interval was 0.05 ms, and no filtering was set for the acquisition process.

2.2.2. Experimental Models with Different Contact States

By excavating the soil around the casing and injecting fluid around the casing in the two cased boreholes in Section 2.2.1, two other experimental models were developed, and combined with the experimental model in Section 2.2.1, three experimental models have been constructed: "contact" model, "free casing" model, and "casing immersed in fluid" model (abbreviated as "fluid immersion" model). For the sake of experimental observation and practical operation, the excavation depth around the casing was 2.2 m, which means that only the first 2.2 m of the casing was falling off from the hole wall in the free casing model. The same is true for the fluid immersion model. Figure 2a–d presents the free casing model and fluid immersion model for PVC cased borehole and steel cased borehole, respectively. Considering the depth of the location where the casing contact state in the borehole changed, 4.0 m was established as the test depth. The distance between measurement points was shortened to 0.5 m. The distance of the percussion sleeper from the borehole remained unchanged, and striking was performed at the same measurement point several times with the signal recorded. 0.05 ms was positioned as the sampling interval, with no filtering set during the acquisition process.

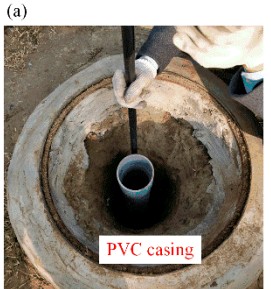
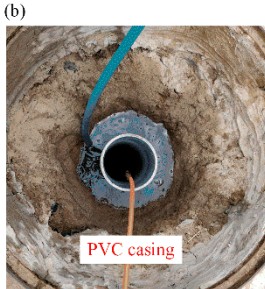
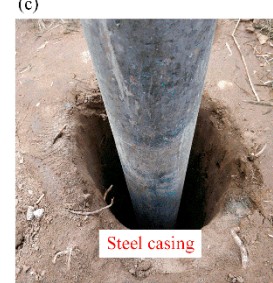
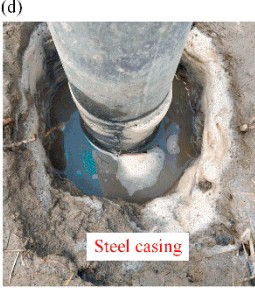

**Figure 2.** Different casing models: (**a**) free casing model of PVC cased borehole, (**b**) casing immersed in fluid model of PVC cased borehole, (**c**) free casing model of steel cased borehole, and (**d**) casing immersed in fluid model of steel cased borehole.

## 3. Numerical Simulation of Wave Velocity Test with Casing Downhole Method

The ABAQUS finite element analysis platform was leveraged to establish a three-dimensional analysis model to simulate and calculate the characteristics of the influence of the cased borehole on the signal obtained through the downhole method wave velocity test. As detailed in Figure 3, the size of the soil in the model was 20 m × 20 m × 30 m, the diameter of the borehole was 0.1 m, and the borehole ran through the whole soil model. The depth of the casing in the borehole from the surface to the bottom of the casing was measured at 20 m. The X–Y plane was shown as the horizontal plane of the actual test, and the center of the borehole was the center of the X–Y plane, which was set as the origin (marked by "O" with co-ordinates (0, 0, 0)); thus, the co-ordinates of the observation point on the casing in the X–Y plane should be (−0.05, 0, 0). In the model, the Z-axis was upward, and the negative Z-axis was the direction of depth increase. ABAQUS infinite element transmission boundaries [21,22] were used around the model and on the bottom surface to mitigate the interference of reflected waves from the model boundaries to the near-field region. The material properties of the infinite element unit were linear elastomer, and the rest of the material parameters was consistent with those of the connected soil unit. An approximate impulse force along the Y-axis was applied to the surface of the model to simulate the excitation source of the sleeper strike in the actual test, and the point of the load action was 2.0 m (marked by "S" with co-ordinates (−2.05, 0, 0)). The observation points on the surface of the casing and the surface of the soil borehole wall in contact with the casing were set along the negative direction of the Z-axis, and the observation points

were located in the X–Y plane; hence, the horizontal Y direction of the observation points was parallel to the direction of the load action.

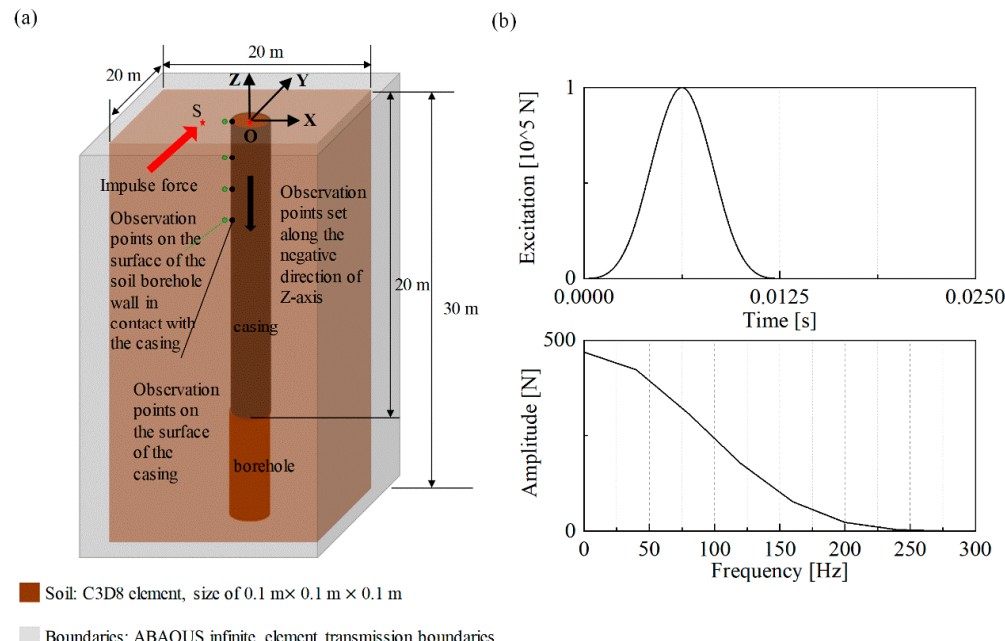

Soil: C3D8 element, size of 0.1 m × 0.1 m × 0.1 m

Boundaries: ABAQUS infinite element transmission boundaries

**Figure 3.** (**a**) Schematic diagram of 3D finite element analysis model, (**b**) approximate impulse concentrated force to simulate surface strike in downhole test.

According to Campanella and Stewart [6], and Ishihara [23], the strain level of the soil in the wave velocity test of the downhole method is in a small range; therefore, the soil was considered as a homogeneous, isotropic linear elastic half-space medium in the simulation of this paper. To minimize the influence of other factors, the soil as well as the casing was set as a single material. Different casing materials were simulated by varying the dielectric shear wave (S-wave) velocity and density of the casing, and by varying the unit contact between the casing and the soil, the contact state in practice was simulated. Given that the soil was in the small strain range under the pulse load, a sound contact state between the casing and the borehole wall was simulated by a common nodal contact between the casing and the soil unit, and a nodal no-contact state was simulated by the casing.

Figure 4 provides a schematic diagram of the different material casing models and the different contact models involved in the simulation. Figure 4a is used to examine the effects of different casing materials on the acquired signals, and to simulate the velocity responses of the measurement points for three conditions: no casing, PVC casing, and steel casing, respectively. Figure 4b,c simulates the effect of free casing between the casing and the borehole wall. In the model of Figure 4b, the free casing occurs in the first 10 m, and the casing is still in contact with the soil at the common node in the last 10 m. Indicated in the model of Figure 4c, the free casing is set in the depth range of 10–15 m, and the casing is still in contact with the soil at the common node in the remaining depth range. Figure 4d presents the new model added to the numerical simulation, in which a soil layer with low dielectric wave velocity is added between the casing and the original soil body to surround the casing. The thickness of the low velocity surround layer is 0.15 m, to simulate the solid medium filling between the casing and the borehole wall, and the three discrete units are in common node contact with each other. The model parameters of the model in Figure 4 are shown in Table 1. Three-dimensional eight-node hexahedral solid units are used to discretize the soil body, and the casing is discretized as a shell unit because its thickness is much smaller than its length, and the degree of freedom in the rotation direction is constrained in the modeling process. The size of soil and casing units and the time step of numerical calculation are set according to the stability criterion of numerical

integration of time domain dynamic display [24], and both the size of the solid unit for discretizing soil and that of the shell unit for discretizing casing are not larger than 0.1 m. The size of unit for discretizing the soil's low velocity layer is not larger than 0.02 m. The soil and casing around 1/4 of the borehole in the X–Y plane is divided into five units. The time step of all models is $1 \times 10^{-6}$ s.

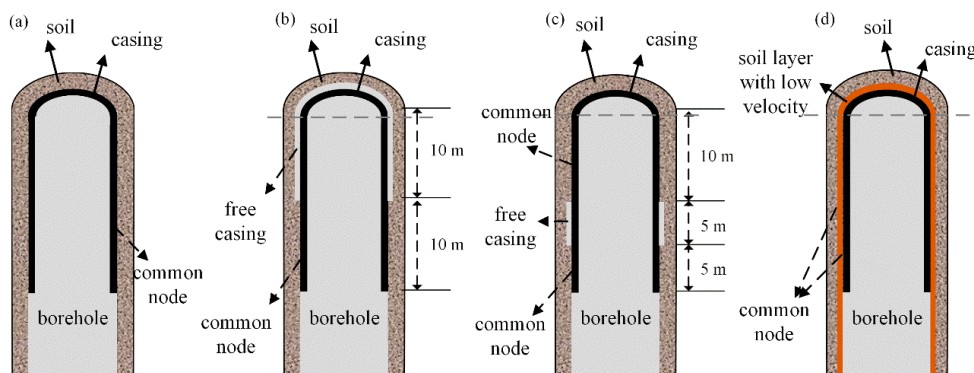

**Figure 4.** Schematic diagram of 4 touching condition model of cased borehole (**a**) cased borehole with different casing material, (**b**) free casing at the first 10 m depth, (**c**) free casing at 15–20 m depth, and (**d**) cased borehole surrounded by a soft soil layer with low shear wave velocity.

**Table 1.** Material parameters of models.

| Model | Simulation Conditions | Soil | | | Casing | | | Contact Mode |
|---|---|---|---|---|---|---|---|---|
| | | Shear Wave Velocity m/s | Poisson's Ratio | Density kg/m³ | Shear Wave Velocity m/s | Poisson's Ratio | Density kg/m³ | |
| Model 1 | Condition 1: free casing | 200 | 0.4 | 1700 | - | - | - | - |
| | Condition 2: PVC casing | 200 | 0.4 | 1700 | 1000 | 0.3 | 1400 | common node |
| | Condition 3: steel casing | 200 | 0.4 | 1700 | 3000 | 0.3 | 7300 | common node |
| Model 2 | Free casing in the depth range of 10 m | 200 | 0.4 | 1700 | 3000 | 0.3 | 7300 | first 10 m: common node post 10 m: free contact |
| Model 3 | Free casing in the depth range of 10–15 m | 200 | 0.4 | 1700 | 3000 | 0.3 | 7300 | 0–10 m: common node 10–15 m: free contact 15–20 m: common node |
| Model 4 | Surrounded by a soil layer with low dielectric wave velocity | 200/ 60 | 0.4/ 0.25 | 1700/ 1400 | 3000 | 0.3 | 7300 | Both contact surfaces are in common node mode |

## 4. Results and Discussion

### 4.1. Effect of Cased Material

Figure 5 displays the horizontal signals and the corresponding spectra acquired in the field experiment for PVC cased borehole and steel cased borehole within 22 m test depth each. In order to lessen the interference of noise, the signals of multiple strikes were recorded and then normalized and superimposed. Except for the signals from the 14 m and 16 m test points, clear peaks were observed at all depths of the PVC cased borehole, and more vibration cycles were noticed at the shallow test points than at the deeper test points

(e.g., 18–22 m test points) as shown in Figure 5a. Through the Fourier variation of the signal obtained from the PVC cased borehole, the signal band was acquired, which ranged from 25 Hz to 100 Hz, similar to the shear band range recorded by other researchers in the downhole method [10,14], and did not change much with the depth of the measurement point. Notwithstanding, it can be seen that the main frequencies of the signals at 4 m, 10 m, and 12 m were higher than those at other depths. These depths were located in the sand layer, and it is assumed that the soil medium characteristics had a greater influence on the signal wave traces and spectral characteristics. The signals observed from the steel cased boreholes were of better quality than those from the PVC cased boreholes, and the signals from the 14 m and 16 m measurement points did not show signal delay, as indicated in Figure 5. Nevertheless, the signal from the shallow measurement points of the steel cased borehole also witnessed more cycles in the wave traces than the signal from the deep measurement points (e.g., 18–24 m measurement points). Shown in Figure 5d, the band range of the steel cased borehole signals was also within the range of 25–100 Hz, close to the band range of the signals obtained from the PVC cased borehole. However, the frequency band of the deeper measurement points obtained from the steel cased borehole was less attenuated around 75 Hz than that from the PVC cased borehole.

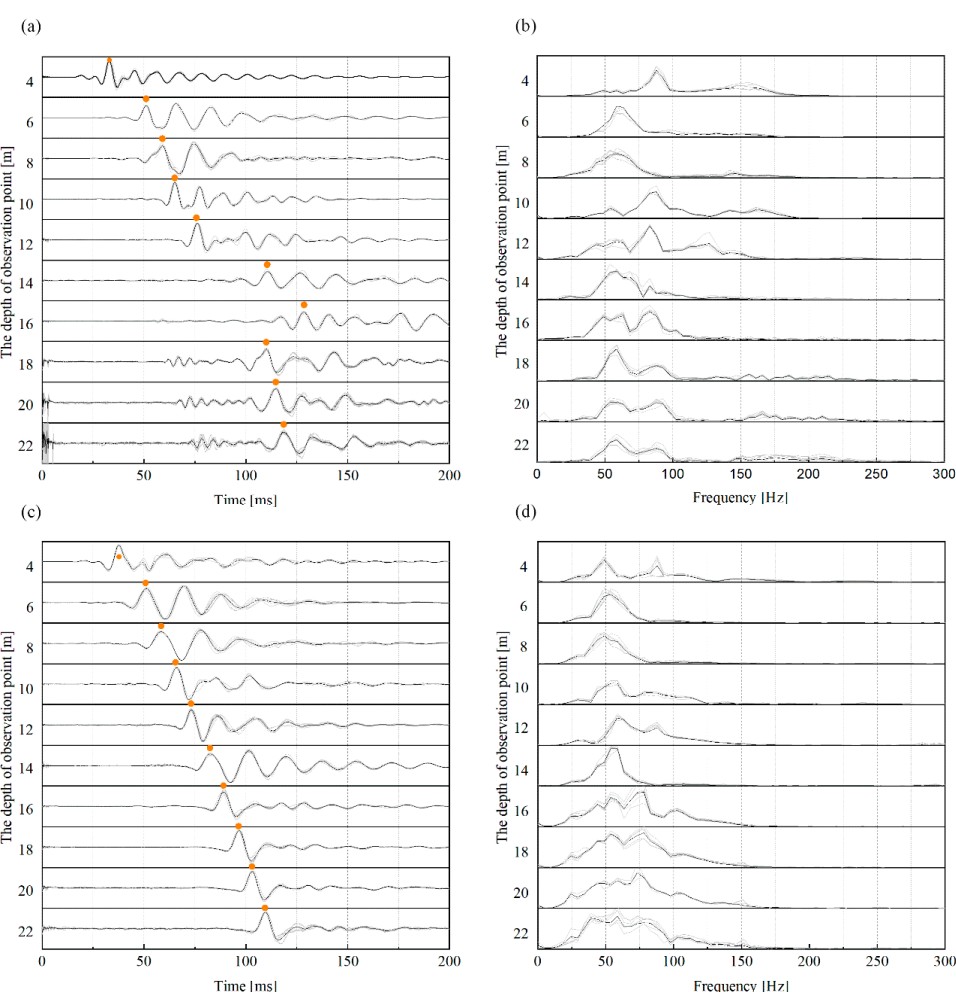

**Figure 5.** (**a**,**b**) Signal traces of horizontal direction of receivers and its corresponding FFT spectra at all testing depths in downhole test with PVC tube, (**c**,**d**) signal traces of horizontal direction of receivers and its corresponding FFT spectra at all testing depths in downhole test with steel tube.

Selecting and calculating the time difference of wave peaks between signals recorded at different depths is a regular approach to calculate shear wave brigades, also known as the peak-to-peak method [6,13]. Then, the shear wave travel distance was calculated based on

the ray path, i.e., the shear wave velocity propagated in a given soil layer could be acquired based on the mathematical relationship between "distance–time–velocity" [6,13,25,26]. Figure 6 exhibits a comparison of the peak times of the signal wave traces recorded in the cased borehole tests of two materials and the shear wave velocities calculated from the signals of the adjacent measurement points. It can be found in Figure 6 that the signal peaks of the two types of cased boreholes were close to each other between 4 m and 12 m. Due to abnormal signal delay at 14 m and 16 m, the signal peaks of the PVC cased boreholes were larger than those of the steel cased boreholes from 14 m onwards. Since the vicinity of 14 m was exactly the partition interface of the site soil layer, and the nature of layer 4 and layer 5 soil was fine sand and powder sand, which could be easily disturbed, thereby leading to the collapse of the hole wall, it is presumed that the delay of PVC casing drilling signal was affected by the bad contact between casing and hole wall. When the depth of the measurement point continued to increase to enter the clay layer, the contact degree between PVC casing and hole wall increased at this time, and the signal gradually returned to normal. For a steel cased borehole with a casing medium wave velocity of up to 2000 m/s, the shear wave velocity was calculated from the recorded signal in the test depth of 4–22 m between 200 m/s and 300 m/s. Similar shear wave velocities were also observed in the PVC cased holes between 6 and 10 m. Due to abnormal signals obtained at 14 m and 16 m, a significant deviation was detected in the shear wave velocity calculated from the other tests of the PVC cased borehole.

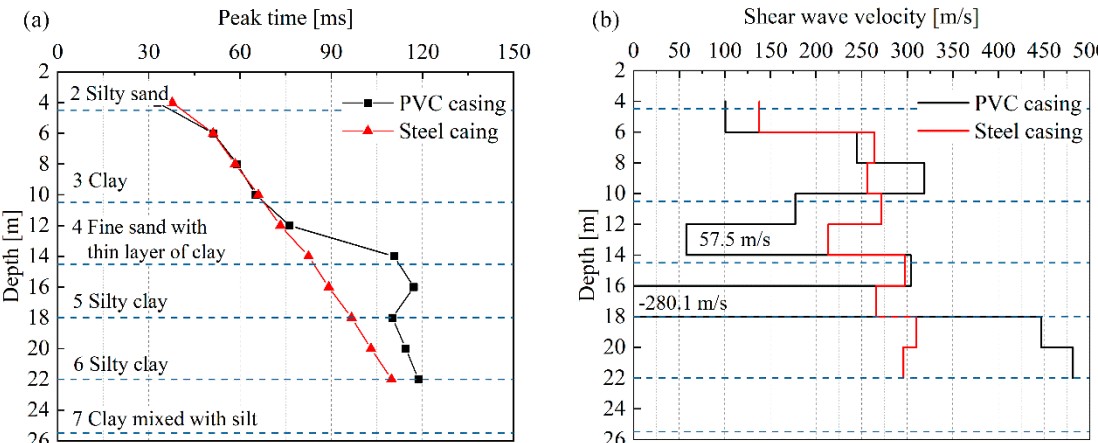

**Figure 6.** (**a**) Comparison of peak time of signals in horizontal direction received in PVC and steel casing downhole test, (**b**) comparison of shear wave velocity calculated through selected peak time based on peak-to-peak method.

As indicated in Figures 5 and 6, the recorded signals from the borehole with casing failed to identify the obvious wave components generated by the casing (possibly due to the small interference components generated by the casing), and the shear wave velocity calculated from the signals was not significantly increased by the casing, whether the shear wave velocity was 700 m/s for the PVC casing or 2500 m/s for the steel casing [7]. Therefore, it is assumed that the borehole with casing can record valuable signals normally, and the quality of the recorded signals in the borehole with casing is more likely to be deteriorated by the unsatisfactory contact between the casing and the borehole wall.

Figure 7 displays the numerical simulation results of the horizontal directional component response. It can be noted that, in the simulation, when the casing and soil are in common contact to simulate the close contact between the casing and the hole wall in the actual test, the amplitude and peak of the velocity response wave traces recorded at the same depth observation point without casing and that with two different media wave velocities are very close. Nonetheless, with the increase in the shear wave velocity of casing media, the amplitude of the response wave traces decreases and the position of the wave peak moves forward continuously. Through comparing the velocity response wave traces

of the observation points at three depth positions, it is found that the amplitude of the response wave traces decreases with the increase in depth and the peak time decreases accordingly when the measurement points are located within the depth range of casing. The peak time difference between adjacent observation points was calculated based on the peak-to-peak method. Figure 7b demonstrates the effects of different casing materials on the calculated shear wave velocity in the numerical simulation. Based on the shear wave velocity test results of the free casing model, it can be observed that the relative error of shear wave velocity calculated by the model with a casing medium wave velocity of 1000 m/s is very small, basically less than 1%. For the casing medium wave velocity model with 3000 m/s, the calculation of shear wave velocity is not affected by the decrease in corresponding wave amplitude and the premise of wave peak time. For example, the relative error of 13–19 m depth is less than 1%. However, it should be noted that the relative error of shear wave velocity calculated by the casing medium wave velocity 3000 m/s model starts at 9 m depth and gradually increases with depth, and this feature still demands further investigation, but the final relative error is only 5%. It can be seen that the relative error of shear wave velocity for both casing models increases significantly near the depth of the bottom of the casing, even up to 13.3% (the model with casing medium wave velocity of 3000 m/s). This suggests that although the casing is in good contact with the soil, the difference in the response wave traces of the observation points between the transition zone of the casing and the soil in the borehole can still generate a major error in the calculated shear wave velocity, especially when the signals of the adjacent observation points are used for calculation.

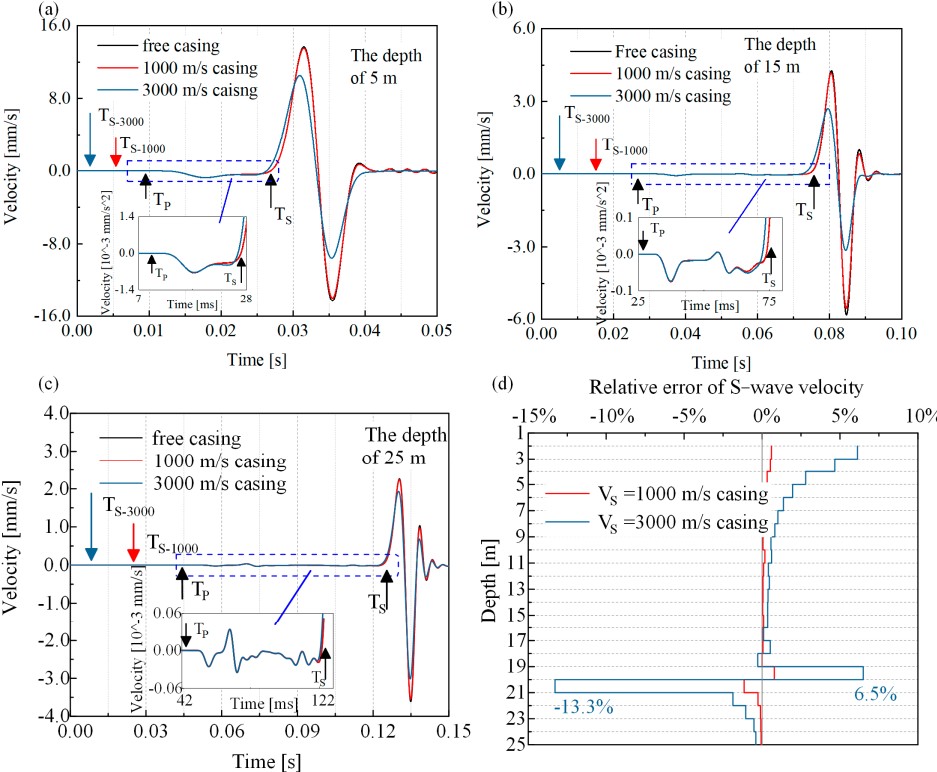

**Figure 7.** Results of numerical simulation of different casing materials models, (**a**) signal traces of horizontal Y-direction component of velocity response at the depth of 5 m, (**b**) signal traces of horizontal Y-direction component of velocity response at the depth of 15 m, (**c**) signal traces of horizontal Y-direction component of velocity response at the depth of 25 m, and (**d**) relative error of S-wave velocity between cased borehole model and non-cased borehole model with shear wave velocity calculation based on the adjacent signals of velocity response using peak-to-peak method.

Based on the numerical simulation results, it can be concluded that the presence of casing does cause errors in the calculated S-wave velocity, and the larger the shear wave velocity of the casing medium, the more significant the errors. However, the actual effect of the presence of the casing on the calculated shear wave velocity is very small, and the only thing that requires attention is the casing transition region. This further reflects that in the state of close contact between the casing and borehole wall, the presence of the casing, and the difference in casing material have little effect on the signal obtained in the borehole, and that the high-quality signal can still be recorded for the calculation of reliable shear wave velocity at this time. In contrast, the variation in the contact state seems to have a more pronounced effect on the recorded signals. For this reason, field experiments are conducted for three contact states and numerical simulations of the downhole method wave velocity tests performed for two of these contact states for better comparative studies.

### 4.2. Effect of Contact State between Casing and Hole Wall

As shown in Figure 2, a free casing model was formed after the soil around the casing was excavated and a "fluid immersion" model after filling the void with fluid. Due to the limitation of manual excavation, only a 2.2 m deep casing could be formed; therefore, the spacing of measurement points was changed to 0.5 m to record enough measurement signals for comparative analysis. Figure 8 presents the signals and spectral information of the 1.5 m measurement points recorded under the three models. By observing Figure 8a, it can be found that the measuring point could still record signals after PVC free casing. The signal arrival time was then delayed compared with the casing contact model, and the signal waveform attenuation slowed down, and there were still obvious up and down vibration wave traces after 50 ms. The frequency band of the signal recorded at the measurement point under the free casing model became significantly narrower, and the main frequency was above 100 Hz, significantly larger. As demonstrated in Figure 8c,d, the wave traces and spectral change characteristics appearing after free casing were especially obvious when the casing material was steel. The test was conducted again after a fluid injection in the void. At this time, the number of cycles of the signal wave traces was similar to that of the contact model, and the peak position of the signal wave traces is different from the delayed position of the casing after dehollowing, but close to the peak position of the contact model again, or even slightly ahead of the contact model. Observing the frequency spectrum of the signal recorded in the water injection model in Figure 8b,d, it was found that the frequency band range increased compared with that in the free casing model and was again close to that in the contact model, especially when the casing material was PVC.

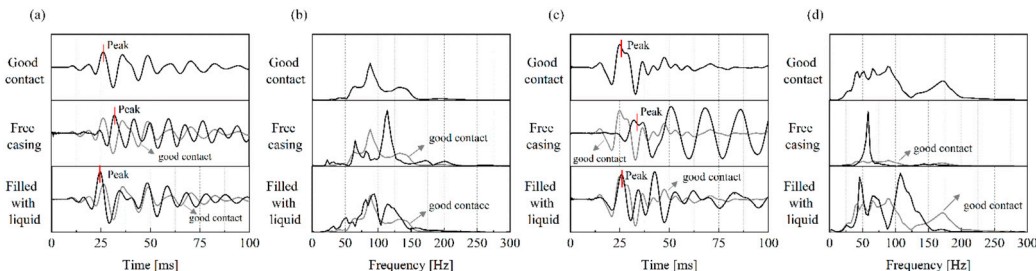

**Figure 8.** Signal traces of horizontal direction and corresponding FFT spectra of 1.5 m receiver in different casing model (**a**,**b**) of PVC cased borehole, (**c**,**d**) of steel cased borehole.

The results in Figure 8 suggest that when the lower part of the casing was still in favorable contact with the hole wall, even if the casing was dehollowed, the signal could still be recorded at the dehollowing location. Based on the signal delay, it can be inferred that the signal propagated upward along the casing through the dehollowing position and then was detected by the measuring point in the dehollowing position. When the casing was surrounded by water, the measured points in the casing could record signals similar to those when the casing was in good contact with the hole wall. This means that in addition

to filling the gap between the casing and the hole wall with sand, pebbles, concrete, and solid objects, the gap filled with liquid could also solve the problem of unfavorable contact between the casing and the hole wall. The liquid could not transmit the shear wave, but since the cased borehole was located in three-dimensional space, shear waves generated by horizontal percussion traveled in a spherical shape, thereby causing the vibration of soil particles. Therefore, more energy was transferred around the casing as shear waves around the liquid and was transferred to the casing in the form of a compression wave, which is then captured by the geophone. Considering that the compressional wave velocity is larger than the shear wave, the signal recorded at the measurement point is advanced, which also explains the advancement of the signal wave crest of the water injection model in Figure 8a,c.

Figure 9 demonstrates the signal and spectrum information of all the measurement points within the test depth of the three contact state models of the PVC cased borehole and the steel cased borehole. It can be seen that in Figure 9a,c, the signal wave traces recorded in the dehollowing area (1.0–2.0 m) and the un-dehollowing area (2.5–4.0 m) differed greatly in the dehollowing model with the excavation surface (2.2 m) as the boundary. Especially when the steel casing was dehollowed, the signal was attenuated very slowly. At the same time, the signal arrival of the measurement point in the dehollowing area was delayed with the increasing shallow depth of the measurement point, shown in Figure 10. This indicates that the wave recorded at the measuring point in the casing dehollowing area propagated along the casing from bottom to top. The recorded signal frequency bands of the measuring points in the casing dehollowing area were narrowed, but still in the range of the soil shear wave (clearly observed in Figure 9c). This means that the signal propagation through the casing to the measurement point still retained the component of the excitation signal propagating through the soil, presumed to be amplified by the influence of the casing when it propagates in the casing, and therefore could be seen in the steel casing, made of more rigid material. Hence, it can be seen that the signal wave traces recorded in the steel casing with greater stiffness had a more obvious feature of increased circulation and a narrower signal band. The shape and spectral characteristics of the recorded signal wave traces in the fluid immersion model were very close to those of the casing contact model. As can be seen from Figure 10, the shear wave velocity between the measurement points calculated using the peak-to-peak method and the linear path of the adjacent measurement points was still different between the contact model and the fluid immersion model, but the relative error of the calculated shear waveform velocity was still within the acceptable range for engineering testing.

Numerical simulation was adopted to make up for the shortcoming of the shallow depth of manual excavation dehollowing in the field test by increasing the depth of the free casing to the first 10 m, and it was also leveraged to calculate the dehollowing condition in the middle region of the casing only, which could be difficult to realize in the field. The modeling process is shown in Section 3. The material parameters of the model were the same as those of the model with good casing contact (Model 1, working condition 3). The numerical results are displayed in Figure 11. For the first 10 m of the casing after dehollowing, the velocity response of the observation point in the dehollowing area showed the characteristics of delayed wave arrival time and increased wave cycle, and the signal arrival time shows an increase with the shallow depth of the observation point, while the spectrum changed from a wide band before dehollowing to two narrower spectral peaks. These changes were similar to the change pattern of the field test results (free casing model). When the dehollowing position occurs in the middle section of the casing, it can be observed from Figure 11c,d that the wave traces of the observation point response arrived earlier than when the casing was in good contact, and the wave peak position was delayed with the increase in the observation point depth, indicating that the wave propagation in the dehollowing area became top-down and propagated at a faster speed than the wave speed of the soil shear wave. The main frequency after the narrowing was larger than that in the first 10 m of the free casing model.

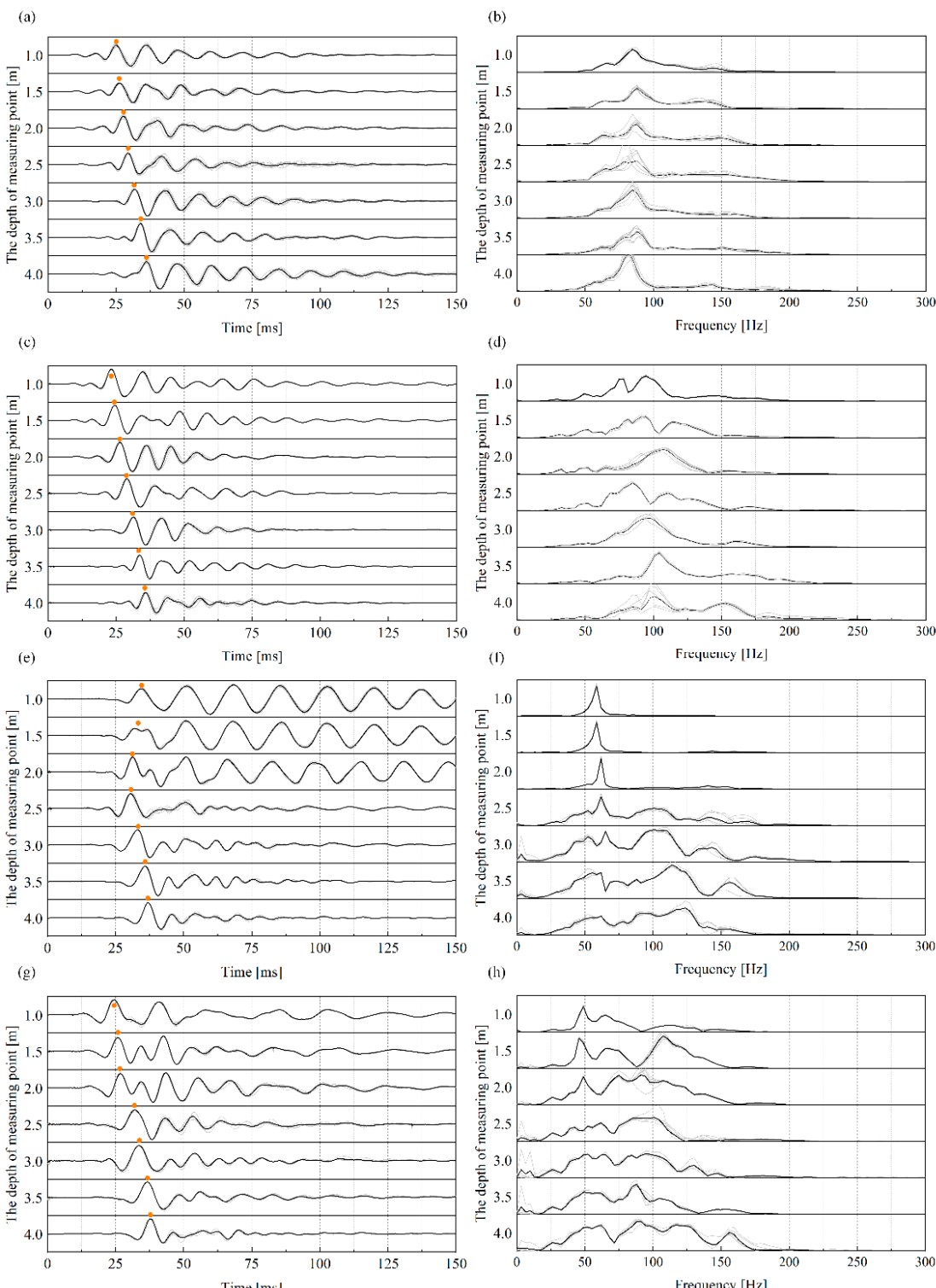

**Figure 9.** (**a**,**b**) Signal traces of horizontal direction and its corresponding FFT spectra of receivers at all testing depths, (**a**,**b**) in free casing model of PVC cased borehole, (**c**,**d**) in casing immersed in fluid model of PVC cased borehole, (**e**,**f**) in free casing model of steel cased borehole, (**g**,**h**) in casing immersed in fluid model of steel cased borehole.

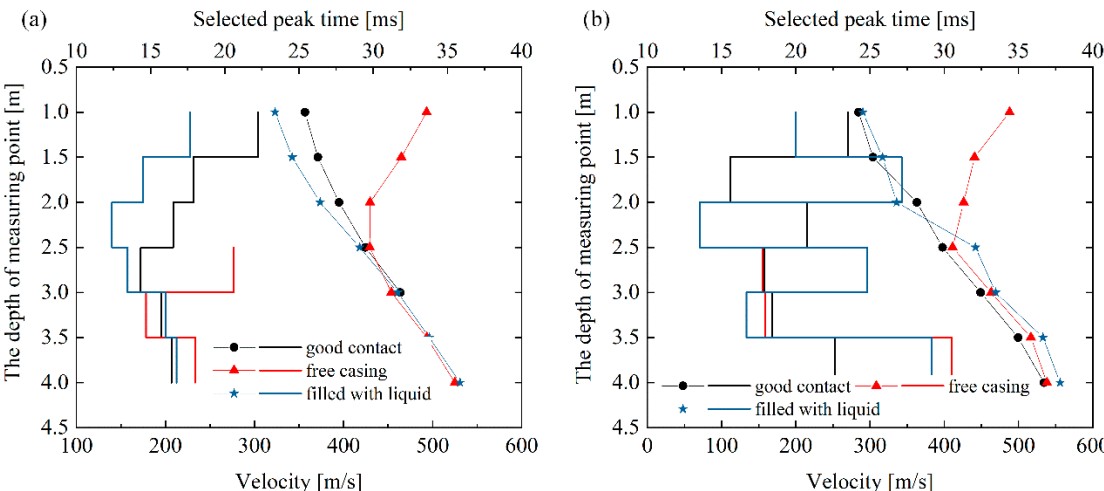

**Figure 10.** Comparison of peak time of signals and shear wave velocity calculated through selected peak time based on peak-to-peak method in three casing model, (**a**) PVC cased borehole, (**b**) steel cased borehole.

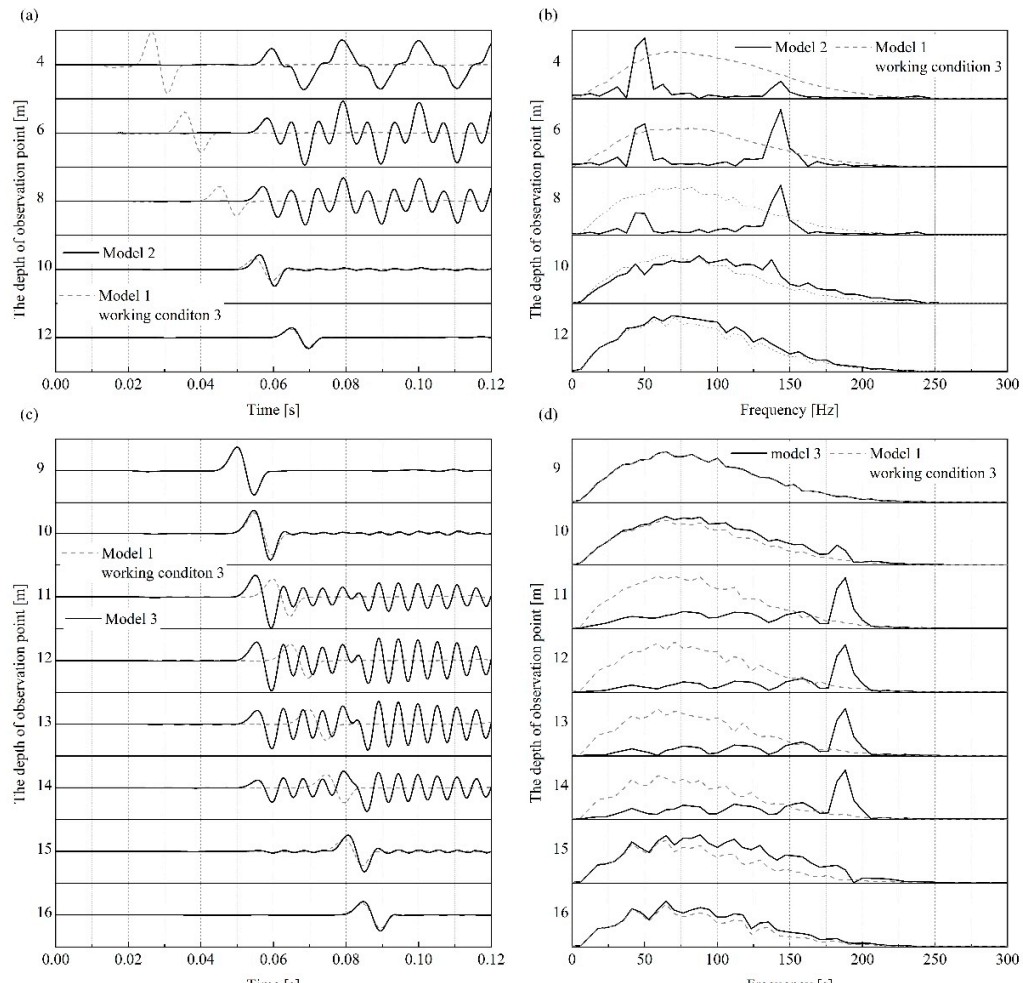

**Figure 11.** Signal traces of horizontal Y-direction component of velocity response and its corresponding FFT spectra from the numerical simulation of case borehole models with different touching conditions, (**a**,**b**) the model with a free casing at the depth of 0–10 m, (**c**,**d**) the model with a free casing at the depth of 10–15 m.

Figure 12 indicates the numerical simulation results of the signals obtained from the measurement points inside the casing after the casing is filled with soft soils with low dielectric wave velocity. Under the modeling conditions of Model 1 and working condition 3, a layer of soil layer with medium shear wave velocity of 60 m/s was added around the casing. The specific modeling parameters are shown in Table 1. As reflected in Figure 12, the newly added medium wave velocity layer caused a decrease in the amplitude and a slight delay in the peak time of the horizontal Y-direction component of the velocity response at the observation point, but the overall wave traces were close to those without the low velocity layer. The results are revealed in Figure 12d, which shows that the calculated shear velocities were similar to those without the low velocity layer at most depths, except for the top and bottom of the casing (19–20 m interval), where the shear velocities were inaccurate compared to those in the good contact model. It can be noted that when the casing was surrounded by a low-velocity soil layer, the signals obtained from the observation points in the borehole were basically the same as those when the casing was surrounded by a preset soil body. However, due to the presence of the low-velocity soil layer, the top of the casing was influenced by the near-field effect, and the error caused by the contact medium conversion of the probe at the bottom of the casing was also magnified.

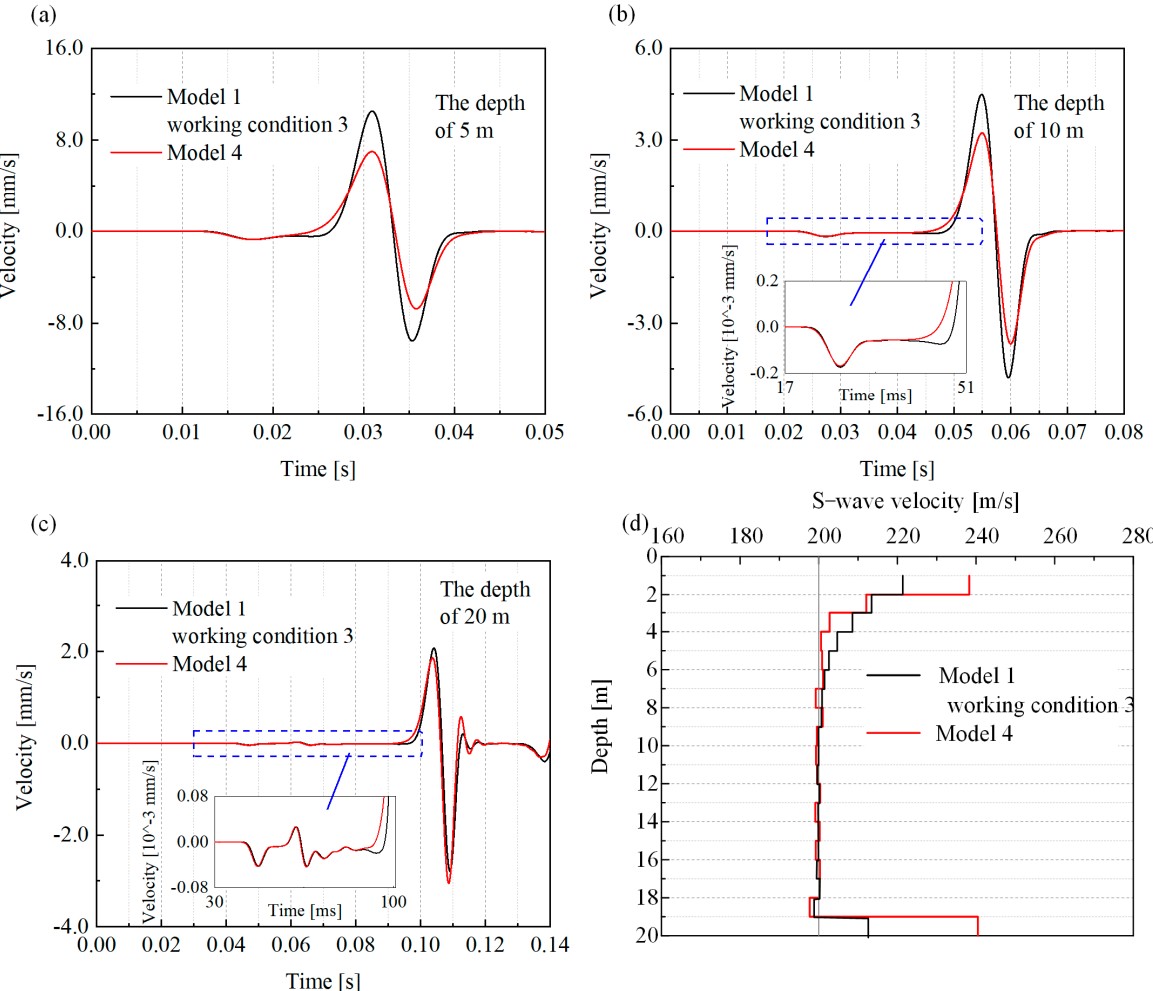

**Figure 12.** Results of numerical simulation of cased model with a casing that is surrounded by low-velocity soil layer, (**a**) signal traces of horizontal Y-direction component of velocity response at the depth of 5 m, (**b**) signal traces of horizontal Y-direction component of velocity response at the depth of 10 m, (**c**) signal traces of horizontal Y-direction component of velocity response at the depth of 20 m, (**d**) shear wave velocity calculation based on the adjacent signals of velocity response using peak-to-peak method.

## 5. Conclusions

In the current research, the wave traces characteristics and spectral characteristics of the signals obtained from the measurement points under different casing materials were compared with the results of 3D finite element numerical simulations of wave velocity tests under the downhole method with casing. For the change of contact form between the borehole and the casing, the influence of the presence of casing dehollowing and the liquid filling in the casing dehollowing area on the signal of the measurement points in the casing was also studied by field experiments and numerical simulations. The main research conclusions are as follows.

(1) Under the condition of desirable contact between the casing and the borehole wall, both PVC cased borehole and steel cased borehole can receive good quality signals in the casing. There are no casing-induced components in the signal wave traces of the measurement point. Both the signal wave traces of different casing materials and the frequency band characteristics are similar. However, due to the existence of the casing, the soil layer prone to collapse and the boundary between soil layers will more easily contribute to the abnormal signal recorded by the measurement points in the casing.

(2) On account of characteristics such as the change in arrival time, the slowdown in wave attenuation, and the narrowing of the main frequency, significant errors in the calculation of shear wave speed will arise. It should be noted that when the dehollowing occurs in a certain area of the casing it will be more complex to identify based on the signals recorded at the dehollowing area, and the shear wave speed calculation error is relatively significant at this time. While dehollowing, the greater the stiffness of the casing material, the greater the impact on the signal of the measurement point.

(3) Fluid injection between the casing and the hole wall can eliminate to a greater extent the error caused by the free casing, but at this time the measurement point signal still contains interference components generated by the casing. However, the calculated shear wave velocity error is within the acceptable range of engineering.

(4) Compared with the presence of casing, the contact state between the casing and the hole wall in the wave velocity test with the casing downhole method has a greater impact on the signal of the measurement point. With a view to acquire better signal of measuring points and eliminate the error of calculating wave velocity, it is an effective approach to, after drilling holes, fill the space between the hole wall and the casing with materials, such as grout and pebble sand.

(5) Two improvements can be made for the further study. One is to apply artificially built site models to accurately control the depth and characteristics of different layers of soil, and better comprehend the impact produced by soil layer differences. The other is to establish a more refined numerical simulation model to simulate the effects of unsaturated soil and the presence of groundwater, as a means to better simulate the actual situations.

**Author Contributions:** S.Y.: Writing—original draft, Visualization. Y.Y.: Conceptualization, Methodology, Supervision, Funding acquisition, Writing—original draft, Writing—review and editing. W.Z.: Methodology, Project administration, Writing—review and editing. J.S.: experiments and experimental data organization. Z.Z.: Methodology, Project administration, Writing—review and editing. All authors have read and agreed to the published version of the manuscript.

**Funding:** This study was supported by the National Key R&D Program of China (No. 2019YFE0115702).

**Institutional Review Board Statement:** Not applicable.

**Informed Consent Statement:** Not applicable.

**Data Availability Statement:** No new data were created.

**Conflicts of Interest:** The authors declare that they have no known competing financial interest or personal relationship that could have appeared to influence the work reported in this paper.

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
