# Peer review of "Influence of Borehole Casing on Received Signals in Downhole Method"

_sustainability, doi:10.3390/su15129805_

Round 1

Reviewer 1 Report

This study mainly focused on the influence of casing on wave velocity testing signal 83 in downhole method. Field experiments on the influence of different materials of casing 84 on the wave velocity test of downhole method were first conducted, and the signal wave 85 characteristics and the usability of test results in the borehole of PVC casing and that of 86 steel casing were analyzed

Reviewer 3 Report

This study aims at clarifying the effects of casing materials and contact conditions on waveform and frequency spectrum of signals. I believe this is a really interesting topic and authors interpreted results soundly, which will be useful for engineers. I would suggest to publish this paper after refining minor problems as follows:

1.       Abstract is not refined and needs further refinement.

2.       In the Introduction, authors should review the latest research results, point out their shortcomings to show the research gap and the particularity of this site, and further highlight the necessity of this study.

3.       The image resolution is low and needs to be supplemented with high resolution images. The Figures should be beautiful.

4.       L.138:“in Section 1.2.1…,and combined with the experimental model in Section 1.2.1”. The section number is wrong.

5.       In Figure 9, the serial number is duplicated.

6.       The paper needs to be checked by the professional English people for the grammar and sentence structure.

7.       Literatures should be collected until 2022 from Web of Science database.

Reviewer 4 Report

This paper used the numerical simulation method to study the influence of borehole casting on received signals in downhole method. The results showed that water injection in the voided area between the casing and soil helped reduce the impact caused by free casing condition. The following comments are for the authors to consider:

(1) Some sentences are quite long. For example, the first sentence in the first paragraph in introduction is too long. These long sentences should be shortened.

(2) The authors conducted literature review in the introduction. However, the shortcomings in previous research or previous references should be highlighted.

(3) The borehole diameter is 117 mm. Please give the reason to use this size.

(4) Figure 1(a) is not clear. Please use a clear version.

(5) In Section 3, the authors mentioned that they used ABAQUS to conduct this simulation. But, I recommend that in the first sentence in Section 3, the authors should mention that in geotechnical engineering, there are many available software, such as FLAC, UDEC or ABAQUS. And the following reference (10.1007/s00603-022-03160-8) can be added to support this. Then, the authors can mention that in this study, the ABAQUS software was used.

(6) A recommendation of further study can be added.

Reviewer 5 Report

The work addresses the influence of well casing materials in downhole tests and the presence of imperfections due to voids. The article is very interesting for the scientific and professional world given the impact of the results in the definition of shear waves. The authors address the issue through in situ tests and numerical simulations. The bibliography appears sufficient for the description of the case in question, an in-depth study of the numerical modeling part would be useful to complete the work.

The paragraph of the description of the experimental tests needs a re-reading and an in-depth analysis of the description of the site and of the tests themselves. The description of numerical modeling is broad and well described. The part of the results is well structured and adequately deals with the comparison between experimental and numerical data.

For the reviewer, the work is accepted but needs additions. In particular, it is recommended in the conclusions to add a critical part of a comment on the precautions to be taken in the technique and what developments research can bring according to the types of soil. Furthermore, a commentary section on the numerical part is necessary in the conclusions.

Minor revisions :

- line 107-113 it refers to Fig 2, maybe figure 1,

- Page 3 insert figure with statigraphy with indication of acquisition depths;

- line 225 (e.g., 18m - 24m .. maybe 18m - 22m;

- Figure 6b negative velocity explain??

- Figure 9 repeated twice a and b

Round 2

Reviewer 4 Report

The authors revised the paper. However, after the sentence of "FLAC, UDEC or Abaqus", the related reference can be added as a support, such as 10.1007/s00603-022-03160-8 (In this paper, the geotechnical simulation software is used to analyse the displacement contour of the roadway excavation). After this correction, I believe this paper can be published.

Author Response

Corresponding modifications have been made in the revised manuscript, please refer to line 94.
